# Histological and Functional Breakdown of the Blood−Brain Barrier in Alzheimer’s Disease: A Multifactorial Intersection

**DOI:** 10.3390/neurolint17100166

**Published:** 2025-10-09

**Authors:** Jordana Mariane Neyra Chauca, Graciela Gaddy Robles Martinez

**Affiliations:** Facultad de Medicina, Universidad Autónoma de Guadalajara, Guadalajara 45129, Jalisco, Mexico; drachela87@gmail.com

**Keywords:** blood–brain barrier, Alzheimer’s disease, degeneration, histological, microglia, inflammation, vascular damage, oxidative stress

## Abstract

**Background:** Alzheimer’s disease (AD) is a multifactorial neurodegenerative disorder characterized by amyloid-β (Aβ) plaques, neurofibrillary tangles, and progressive cognitive decline. Recent evidence has highlighted the role of blood–brain barrier (BBB) dysfunction in the early stages of AD pathology. **Objective:** We sought to explore the histological structure and physiological function of the blood–brain barrier, and to identify the shared pathological mechanisms between BBB disruption and Alzheimer’s disease progression. **Methods:** This narrative review was conducted through a comprehensive search of peer-reviewed literature from 1997 to 2024, using databases such as PubMed, Elsevier, Scopus, and Google Scholar. **Results:** Multiple histological and cellular components—including endothelial cells, pericytes, astrocytes, and tight junctions—contribute to BBB integrity. The breakdown of this barrier in AD is associated with chronic inflammation, oxidative stress, vascular injury, pericyte degeneration, astrocyte polarity loss, and dysfunction of nutrient transport systems like Glucose Transporter Type 1 (GLUT1). These alterations promote neuroinflammation, amyloid-β (Aβ) accumulation, and progressive neuronal damage. **Conclusions:** BBB dysfunction is not merely a consequence of AD but may act as an early and active driver of its pathogenesis. Understanding the mechanisms of BBB breakdown can lead to early diagnostic markers and novel therapeutic strategies aimed at preserving or restoring barrier integrity in Alzheimer’s disease.

## 1. Introduction

Alzheimer’s disease (AD) is the most common cause of dementia worldwide, currently affecting more than 150 million people, with prevalence expected to rise sharply in aging populations [1]. Traditionally characterized by amyloid-β (Aβ) plaques and tau neurofibrillary tangles, AD is now increasingly recognized as a disorder that also involves vascular and immune dysfunctions. In particular, disruption of the blood–brain barrier (BBB)—a dynamic interface regulating molecular transport and neurovascular signaling—has emerged as an early and potentially causal event in AD pathogenesis. Evidence indicates that a weakened BBB facilitates the infiltration of circulating Aβ into the brain, exacerbates neuroinflammation, and accelerates neuronal injury [2,3].

Although both AD and BBB dysfunction have been extensively reviewed [4,5,6], the novelty of the present article lies in examining their multifactorial intersection. By integrating histological, functional, and mechanistic perspectives, this review aims to clarify how BBB breakdown actively contributes to AD progression and to highlight its relevance as a diagnostic and therapeutic target [7].

Mounting evidence shows that BBB dysfunction impairs the clearance of Aβ [7,8], fosters tau hyperphosphorylation and propagation through sustained neuroinflammation [9,10], and induces vascular hypoperfusion and oxidative stress, which exacerbate synaptic loss and neuronal death [11,12]. These interrelated processes illustrate that BBB disruption does not occur in isolation but constitutes a converging mechanism that integrates vascular, metabolic, and neuroimmune dysfunctions. By emphasizing these overlapping pathways, the present review highlights how BBB breakdown is emerging as a pivotal factor linking diverse aspects of AD pathology.

In addition, recent studies have brought renewed attention to unresolved controversies [13,14,15]: Does BBB breakdown precede amyloid deposition, or does it develop in parallel? Is it an early trigger of cognitive decline or a secondary amplifier of neurodegeneration? Addressing these questions is critical, as they determine whether the BBB should be considered not only a biomarker of disease progression but also a direct therapeutic target on par with Aβ and tau.

The concept of the BBB was first introduced by Paul Ehrlich in 1885, who observed that certain dyes failed to stain the brain despite circulating systemically. This foundational discovery underscored the unique protective properties of the brain’s vasculature and laid the groundwork for subsequent research into BBB structure and function, particularly its relevance in neurodegenerative diseases such as AD.

Alzheimer’s disease currently affects approximately 150 million people worldwide, according to the World Health Organization (WHO) [1]. The social and economic burden is immense, and with global aging trends, this prevalence is expected to rise sharply in coming decades. A weakening of the BBB may allow increased infiltration of circulating Aβ into the brain, complementing endogenous production and accelerating neuronal deterioration [2,3]. This perspective not only reframes the BBB as a central player in AD but also underscores the urgent need to integrate vascular biology into the broader framework of neurodegeneration.

## 2. Method

This narrative review was conducted in accordance with the Scale for the Assessment of Narrative Review Articles (SANRA) guidelines to ensure methodological clarity and academic rigor. A comprehensive literature search was performed in PubMed, Scopus, and Web of Science as primary databases, with Google Scholar used exclusively for backward and forward citation tracking. The time window was set between January 1997 and December 2024. Boolean combinations of the following keywords were applied: “blood–brain barrier,” “Alzheimer’s disease,” “BBB permeability,” “neuroinflammation,” “pericyte,” “astrocyte,” and “tight junction.”

Articles were eligible if they addressed histological, functional, or pathological aspects of the BBB in the context of AD. Both human and relevant animal model studies were included. Exclusion criteria were applied to case reports, conference abstracts, non–peer-reviewed material, and articles focusing on unrelated neurological conditions. Two independent reviewers screened titles and abstracts, followed by a full-text evaluation. Any disagreements were resolved by consensus.

The initial search retrieved more than 23,000 records. After removal of duplicates, application of time-window, language filters (English/Spanish), and restriction to peer-reviewed original and review articles, 130 references were retained. Of these, 58 articles fulfilled all inclusion criteria and were ultimately incorporated into the qualitative synthesis. To improve clarity, methodological details previously placed after the results have been repositioned here, in line with scientific writing conventions.

## 3. Association Between Blood–Brain Barrier Dysfunction and Alzheimer’s Disease

### 3.1. Structure of the Blond-Brain Barrier (BBB)

#### 3.1.1. Blood–Brain Barrier (BBB) in the Neurovascular Unit

The blood–brain barrier (BBB) is a complex, semi-permeable, and highly specialized interface composed of multiple cell types (Figure 1). Its primary function is to preserve the stability of the brain’s internal environment by restricting the passage of potentially harmful substances from the bloodstream into the central nervous system (CNS) [16,17]. This selective barrier poses a major challenge for the treatment of neurological diseases, as it also hinders the delivery of many therapeutic agents into brain tissue, including those intended for neurodegenerative conditions and brain malignancies [18].

Because neuronal cells are located extremely close to brain capillaries—at distances typically under 25 μm—the BBB represents a strategic and efficient route for targeted drug delivery, as opposed to longer and less effective systemic routes [16,17]. This proximity has encouraged scientific exploration into novel methods that can transiently modulate BBB permeability or bypass its restrictions altogether. Researchers are also focusing on how various delivery systems interact with the BBB and what structural features govern its selective permeability [18,19,20].

The structural integrity of the BBB depends on several key proteins that form intercellular junctions. Among them, claudins, occludins, and junctional adhesion molecules play a central role in sealing the endothelial lining. These are accompanied by cytoplasmic scaffold proteins such as Zonula Occludens (ZO-1), ZO-2, ZO-3, and cingulin, all of which interact with the actin cytoskeleton to support barrier function [16,17]. Claudins—particularly claudin-1 and claudin-5—along with occludin, are essential components of tight junctions. Occludin, a phosphoprotein, works in concert with other adhesion molecules, though the full extent of their function within the BBB remains under investigation [21]. Overall, the BBB is sustained by the coordinated efforts of endothelial cells, astrocytes, pericytes, and the complex junctional systems they form [22,23].

#### 3.1.2. Endothelial Cells

Endothelial cells are not passive structural elements but dynamic regulators of the blood–brain barrier (BBB). They control the selective passage of ions, nutrients, and signaling molecules, and they actively respond to systemic and neural cues that modulate barrier permeability [18,21]. Through cross-talk with pericytes and astrocytic endfeet, they coordinate vascular stability and contribute to neurovascular coupling [23].

Endothelial cells form the primary structural layer of the blood–brain barrier (BBB), characterized by the presence of tight junctions composed mainly of claudin-5, occludin, and junctional adhesion molecules. These junctions regulate paracellular permeability, while the low rate of transcytosis and specific transport systems such as Glucose Transporter Type 1 (GLUT1) and P-glycoprotein maintain strict molecular selectivity [12,13,14]. These specialized features are critical for preserving homeostasis in the central nervous system (CNS) [21].

#### 3.1.3. Astrocytes

Astrocytes interact with endothelial cells via their perivascular endfeet, releasing signals like Sonic hedgehog and angiopoietins to support BBB maintenance. They contribute to the expression of tight junction proteins, ionic balance, and water homeostasis through aquaporin-4 channels [24,25]. Dysfunctional astrocytic signaling is involved in BBB breakdown in various neurological disorders, including AD [26].

#### 3.1.4. Pericytes

Pericytes, embedded within the basal lamina, are critical regulators of blood–brain barrier (BBB) integrity. They provide structural support to endothelial cells, modulate vascular stability, and control capillary blood flow through signaling pathways mediated by the platelet-derived growth factor receptor beta (PDGFRβ) [27,28]. Experimental and clinical evidence has shown that pericyte loss or dysfunction leads to increased BBB permeability, cerebral hypoperfusion, and the accumulation of neurotoxic proteins such as Aβ, contributing to cognitive decline in Alzheimer’s disease (AD) [8,29,30]. In addition to their structural role, pericytes actively regulate endothelial tight junction integrity and suppress excessive transcytosis, mechanisms essential for maintaining barrier selectivity [5,12]. Importantly, PDGFRβ refers to the membrane-bound receptor essential for vascular stability, whereas a soluble form (sPDGFRβ), detectable in cerebrospinal fluid and plasma, has emerged as a biomarker of pericyte injury and correlates with BBB breakdown and AD-related neurodegeneration [8,29]. These findings highlight that pericyte degeneration is not an isolated event, but a central mechanism linking vascular dysfunction to the pathophysiology of AD.

#### 3.1.5. Tight Junctions

Tight junctions are essential for establishing the selective permeability of the BBB. These junctional complexes seal the paracellular space between endothelial cells, thereby limiting the diffusion of hydrophilic and high-molecular-weight substances. They act as both “gates”—regulating substance passage—and “fences”—preserving membrane polarity. Tight junctions are composed of transmembrane proteins such as claudins and occludins, which connect adjacent cells and are anchored to the cytoskeleton via adaptor proteins like ZO-1. Additional elements such as tricellulin, lipolysis-stimulated lipoprotein receptors, and adhesion molecules contribute to the maturation and functionality of tight junctions [31,32]. Kinases and other regulatory proteins further influence tight junction permeability, and any downregulation in their expression can lead to BBB disruption.

#### 3.1.6. Basement Membranes

The basement membranes (BMs) provide structural and biochemical support to the neurovascular unit, surrounding endothelial cells, pericytes, and astrocytic endfeet as specialized extracellular matrix layers. They are mainly composed of laminins, collagen type IV, nidogen, and heparan sulfate proteoglycans, which together form a scaffold that stabilizes the blood–brain barrier (BBB) architecture [18,20]. In addition to offering mechanical strength, BMs regulate critical processes such as cell adhesion, migration, and signaling pathways that maintain vascular integrity. They also act as a selective filter, modulating the trafficking of molecules and immune cells between the periphery and the central nervous system [21]. Importantly, disruption of the basement membranes—through degradation of laminins or collagen IV—has been associated with BBB breakdown, perivascular inflammation, and impaired clearance of toxic proteins such as Aβ in neurodegenerative disorders [33].

#### 3.1.7. Adherens Junctions

Adherens junctions complement tight junctions by maintaining endothelial cohesion and supporting structural integrity. These microdomains are composed of vascular endothelial (VE)-cadherin and cytoplasmic catenins, which link transmembrane adhesion complexes to the actin cytoskeleton. VE-cadherin, in particular, mediates homophilic interactions between endothelial cells and stabilizes cell–cell adhesion [7,20]. Catenins, p120 proteins, and placoproteins act as scaffolds that bridge adherens and tight junctions via ZO-1. Other molecules such as PECAM-1, CD99, and nectins also participate in adherens junction organization. Disruption of these complexes compromises barrier integrity and may lead to vascular leakage [18,33].

### 3.2. Function of the Blood–Brain Barrier

The blood–brain barrier (BBB) exerts multiple essential functions that are critical for maintaining central nervous system homeostasis. Its tightly organized endothelial layer protects the brain by preventing the entry of circulating toxins and non-essential molecules, while allowing oxygen, glucose, amino acids, and other vital nutrients to reach the parenchyma [18,20]. This selectivity is achieved through specialized transport mechanisms, including facilitated diffusion—for example, glucose uptake via GLUT1—and ATP-dependent active transport systems that ensure a precise metabolic supply to neural tissue [4,21,27]. In addition, the BBB is not simply a passive filter; it functions as a metabolically active interface capable of modifying or degrading substances during their passage between the blood and nervous tissue, thereby tightly regulating the cerebral microenvironment [23,34].

### 3.3. Transport Mechanisms Through the Blood–Brain Barrier

Brain capillary endothelial cells differ significantly from their peripheral counterparts due to the abundance of tight junctions, scarcity of vesicular transport, and elevated mitochondrial density. These tight junctions not only restrict paracellular permeability but also diversify the membrane compositions on opposite sides of the endothelial layer [4]. To overcome the restrictive nature of the BBB, various specialized transport systems have evolved to facilitate the controlled entry of molecules into the central nervous system [4,23,27].

#### 3.3.1. Passive Diffusion

Small molecules may cross the BBB through either paracellular gaps or by transcellular routes. However, only a limited range of hydrophilic molecules can diffuse passively between tight junctions. In contrast, lipophilic compounds—such as ethanol and steroid hormones—can dissolve within the lipid bilayer of endothelial membranes and diffuse directly into the brain. For most essential nutrients, such as glucose and amino acids, passive diffusion is insufficient, necessitating more selective transport mechanisms [27].

#### 3.3.2. Carrier-Mediated Transport

Integral membrane transporters enable specific nutrients and metabolites to cross the BBB efficiently. These proteins recognize and bind to their respective solutes—such as glucose or amino acids—and undergo structural shifts that ferry the molecules from regions of higher to lower concentration. When movement against a concentration gradient is required, energy in the form of ATP is employed, especially for charged or polar molecules [7,35,36].

#### 3.3.3. Active Efflux Mechanisms

Efflux transporters actively remove both endogenous toxins and exogenous compounds, including therapeutic agents, from the brain into the bloodstream. A critical group in this function is the ATP-binding cassette (ABC) transporter family, which significantly influences drug pharmacokinetics within the central nervous system. While these pumps protect the brain from harmful agents, they also present a barrier to effective pharmacological treatment of neurological conditions [9,29,33].

#### 3.3.4. Receptor-Mediated Transcytosis (RMT)

Receptor-mediated transcytosis enables selective transport of macromolecules via interaction with specific endothelial cell surface receptors. Molecules such as insulin, transferrin, and apolipoproteins bind to their respective receptors located in clathrin-coated pits of the luminal membrane. These invaginations form vesicles that internalize the ligand–receptor complex. After acidification within endosomes, the ligand may be released and transported across to the opposite side of the endothelial cell layer [9,29,33].

#### 3.3.5. Adsorptive-Mediated Transcytosis (AMT)

Adsorptive-mediated transcytosis, often referred to as non-specific pinocytosis, involves the electrostatic interaction between cationic molecules and the negatively charged luminal surface of endothelial cells. Following this interaction, the bound molecule is internalized and transported through the endothelial cytoplasm, supported by the abundant mitochondria found within these cells, and eventually exocytosed on the abluminal side [9,27,33].

#### 3.3.6. Cell-Mediated Transcytosis (CMT)

Cell-mediated transcytosis operates through the so-called “Trojan horse” mechanism, in which immune cells—such as monocytes or macrophages—carry various molecules across the BBB [27,29,33]. This method enables the transport of a broader range of substances, including those that cannot otherwise traverse the barrier. For instance, HIV-infected monocytes exploit this mechanism to infiltrate the brain. Emerging evidence also supports the use of CMT in therapeutic delivery, particularly in targeting drugs to the CNS [9,29,33].

### 3.4. Circumventricular Organs and the BBB

It is worth mentioning that there are also areas in the brain whose capillaries have fenestrated endothelium; that is, they do not have a BBB, with consequent free exchange of molecules between the blood and neurons. These sites are mainly located around the ventricular cavities, and they are therefore called circumventricular organs. These structures include the lamina terminalis vascularis organ (LTO), subfornical organ (SFO), subcommissural organ (SCO), median eminence, pineal gland, neurohypophysis, and area postrema (AP). It is worth mentioning that choroid plexuses have been excluded from the list of circumventricular organs, although the reason for this is not clearly explained [33,35,37].

### 3.5. Mechanisms of Blood–Brain Barrier Breakdown

Multiple interrelated processes drive BBB disruption in Alzheimer’s disease, each of which builds on the structural vulnerabilities described in earlier sections. Inflammatory mediators such as interleukin-1 beta (IL-1β) and tumor necrosis factor alpha (TNF-α) alter the integrity of tight and adherens junctions, weakening the paracellular barrier formed by endothelial cells [1,7,21]. These two cytokines are highlighted because they are the most consistently reported and mechanistically characterized mediators of BBB dysfunction in AD, exerting effects on endothelial signaling, junctional disassembly, leukocyte adhesion, and microglia–endothelium cross-talk [1,21,29,38,39]. Nevertheless, other cytokines—including interleukin-6 (IL-6) and interleukin-10 (IL-10)—also contribute to these processes and have been incorporated here as representative examples [29,33,39]. Vascular injury, particularly ischemic and hypoxic insults, further destabilizes the BBB by inducing VEGF and nitric oxide signaling, which increase endothelial permeability and promote leakage [18,34]. Oxidative stress exacerbates these effects by directly damaging endothelial cells, junctional proteins, and basement membrane components, thereby amplifying barrier failure [4,33,40]. In parallel, microglial activation at sites of vascular injury releases cytokines and matrix metalloproteinases that facilitate leukocyte infiltration, linking immune dysregulation to neurovascular damage [22,39]. Pericyte degeneration, previously discussed as a key regulator of vascular stability, compounds these mechanisms by reducing capillary support and contributing to hypoperfusion in vulnerable regions such as the hippocampus [8,28]. Similarly, loss of astrocytic polarity disrupts metabolic and ionic homeostasis, weakening neurovascular coupling and further aggravating barrier dysfunction [2,6,20,33]. Taken together, these converging mechanisms illustrate that BBB breakdown is not the result of a single pathway but the outcome of cumulative and synergistic insults that integrate vascular, inflammatory, and metabolic disturbances in AD.

### 3.6. Blood–Brain Barrier Disruption and Its Role in Neurological Disorders

Growing evidence indicates that the breakdown of the blood–brain barrier (BBB) in Alzheimer’s disease (AD) is not merely a secondary outcome but a central pathological process that precedes and contributes to neurodegeneration [25,27,41,42]. The BBB is a complex, dynamic interface that regulates molecular transport, maintains ionic homeostasis, and mediates neuroimmune interactions through a tightly regulated structure composed of endothelial cells, pericytes, astrocytic endfeet, and basement membrane components [18,20,22]. In AD, the deterioration of this structure is multifactorial and manifests both histologically and functionally in ways that critically disrupt central nervous system homeostasis [3,4,7,29]. Rather than treating BBB dysfunction and AD pathology as parallel or independent phenomena, recent integrative perspectives highlight their mutual reinforcement, positioning BBB breakdown as a central driver of AD pathogenesis [2,9,35].

Histologically, AD is associated with the disassembly of tight and adherens junctions among endothelial cells, leading to increased paracellular permeability and vascular leakage [20,25,33]. This is compounded by pericyte degeneration, which compromises capillary integrity and promotes hypoperfusion in regions such as the hippocampus—an area highly susceptible to early AD pathology [8,28]. Astrocytes, essential for regulating BBB permeability, also lose their perivascular polarity and metabolic coupling with neurons and endothelial cells, which weakens their capacity to maintain BBB integrity [6,20,33]. Furthermore, microglial activation near compromised vessels exacerbates barrier disruption through the release of pro-inflammatory cytokines and matrix metalloproteinases, accelerating basal lamina degradation [36,39].

Functionally, the BBB in AD displays a reduced expression of transporters such as GLUT1, P-glycoprotein (P-gp), and low-density lipoprotein receptor-related protein 1 (LRP1), which are essential for nutrient delivery and Aβ clearance [21,23,29]. The impaired clearance of Aβ across the BBB promotes its accumulation in the brain parenchyma, forming extracellular plaques that further destabilize the neurovascular unit [3,4,7]. In parallel, increased BBB permeability allows the entry of peripheral immune cells and neurotoxic molecules into the brain, amplifying inflammation and oxidative stress. Circulating cytokines such as IL-1β, IL-6, and TNF-α cross the damaged barrier and perpetuate microglial activation, which in turn promotes synaptic dysfunction and neuronal injury [1,29,43].

Neuroimaging studies in cognitively normal individuals at risk of AD have revealed BBB leakage in the hippocampus prior to the appearance of amyloid plaques, suggesting that BBB disruption may act as an early biomarker and pathogenic trigger in AD [27,41,42]. Transcriptomic and immunohistochemical analyses of human brain tissue further confirm alterations in endothelial and mural cell gene expression associated with angiogenesis, mitochondrial function, and tight junction assembly in AD [39,40,44,45,46]. These changes contribute to vascular dysregulation, impaired cerebral perfusion, and chronic neuroinflammation, reinforcing the role of the BBB as a central player in AD pathogenesis.

Rather than treating BBB dysfunction and Alzheimer’s pathology as parallel or independent phenomena, an integrative view highlights their mutual reinforcement [2,9,35,44]. The convergence of these processes suggests that AD should be reconsidered, at least in part, as a neurovascular disorder. Therapeutic approaches aimed at preserving BBB integrity—by stabilizing endothelial function, preventing pericyte loss, modulating inflammatory signaling, or enhancing transport systems—may therefore offer promising avenues for early intervention in AD.

While Alzheimer’s disease remains the primary focus of this section, BBB disruption also plays a role in other major neurological disorders. In multiple sclerosis, barrier breakdown facilitates the infiltration of lymphocytes and macrophages into the CNS, promoting demyelinating lesions [20,33]. Histopathological studies show perivascular cuffing, fibrinogen leakage, and chronic microglial activation in MS plaques, indicating that BBB leakage is an early and sustained event in lesion development.

In ischemic stroke, endothelial damage and tight junction degradation contribute to vasogenic edema and secondary injury [40,44]. Reperfusion can further aggravate barrier dysfunction through oxidative stress and matrix metalloproteinase activation, which disrupts the basement membrane and enhances neuronal loss.

In Parkinson’s disease, regional BBB impairment may allow the entry of neurotoxins and exacerbate dopaminergic vulnerability [29]. Evidence from animal models demonstrates that reduced P-glycoprotein activity and increased permeability in the substantia nigra contribute to selective degeneration of dopaminergic neurons.

In epilepsy, chronic BBB dysfunction has been linked to glutamate excitotoxicity, albumin extravasation, and increased seizure susceptibility [39]. Experimental studies confirm that serum albumin entering the parenchyma activates astrocytes and TGF-β signaling, leading to hyperexcitability and lowering the seizure threshold.

Collectively, these examples highlight BBB breakdown as a unifying pathological feature across multiple neurological disorders, even though the downstream consequences vary by disease context.

### 3.7. The Blood–Brain Barrier and Alzheimer’s Disease

Although Alzheimer’s disease was long considered a disease exclusive to the elderly, it has begun to be detected in very young patients. It has been proven that addictions, such as alcoholism, smoking, and drug addiction, can cause its early onset, since certain substances enter the bloodstream and, due to their high toxicity, alter the behavior of the BBB, reach neurons, and alter them.

To corroborate this, contrast-enhanced brain MRI scans were performed, and the images showed that the blood–brain barrier that protects the brain developed leaks with age. These leaks begin in the hippocampus, an important learning and memory center that is damaged in Alzheimer’s disease. Postmortem examination of the brains of patients with Alzheimer’s disease has also revealed damage to the blood–brain barrier [41,47].

Emerging evidence from human studies suggests that BBB breakdown may indeed precede amyloid pathology in Alzheimer’s disease. For instance, neuroimaging studies using contrast-enhanced MRI have demonstrated early BBB leakage in the hippocampus of cognitively normal elderly individuals who are at risk for AD, prior to detectable amyloid deposition. Additionally, elevated levels of BBB breakdown markers—such as soluble platelet-derived growth factor receptor beta (sPDGFRβ)—have been found in the cerebrospinal fluid of preclinical AD patients. These findings support the hypothesis that BBB disruption is not merely secondary to amyloid accumulation but may represent an initiating event in AD pathogenesis [3,9].

## 4. Comparison Between the Causes of Alzheimer’s Disease and Blood–Brain Barrier Impairment

### 4.1. Function of Microglia

#### 4.1.1. Blood–Brain Barrier

Under normal conditions, microglial activity is tightly regulated by the intact blood–brain barrier (BBB), which prevents direct exposure to circulating blood components. However, in the setting of acute injury or neurodegeneration, BBB disruption allows plasma-derived molecules to infiltrate the brain parenchyma, activating microglial responses. To investigate the specific impact of circulating plasma factors, researchers introduced plasma from wild-type mice into the corpus callosum of recipient animals. The result was a significant shift in microglial gene expression, particularly in pathways associated with cytoskeletal reorganization, oxidative phosphorylation, stress response, and transcriptional control.

Interestingly, when plasma from fibrinogen alpha chain-deficient mice (Fga-/-) was used, a marked downregulation of genes related to oxidative stress (e.g., *Hmox1*, *Cox7a2*, *Slc25a5*) and disease-associated microglial profiles (e.g., *Ccl12*, *Rps8*, *Rpl35*, *Atp5e*, *Psmd2*, *Tubb5*) was observed. These findings point to fibrinogen—a key coagulation protein—as a potent activator of microglia, particularly in the context of cerebrovascular events such as stroke or minor hemorrhage [30,36].

Red blood cells (RBCs), when exposed to oxidative insults, can also provoke microglial activation and impair cerebral microcirculation. In experiments using Tie2-GFP transgenic mice—which express fluorescent markers in endothelial cells—oxidatively stressed RBCs (labeled with PKH26 and treated with t-butyl hydroperoxide) were injected to simulate a redox challenge. Compared with controls injected with phosphate-buffered saline (PBS), these animals showed increased vascular stasis and reduced cerebral blood flow within 1 to 24 h post-injection, as visualized through two-photon imaging [48].

Additionally, perivascular microglia—distinct from resident parenchymal microglia—derive from bone marrow precursors and localize to the vasculature. These macrophage-like cells are strategically positioned to mediate immune surveillance and may play a central role in BBB dysfunction during autoimmune and inflammatory diseases, particularly those involving leukocyte transmigration across the endothelium [7].

These BBB alterations not only impair vascular homeostasis but also accelerate Alzheimer’s disease progression by facilitating amyloid-β accumulation, enhancing neuroinflammation, and reducing cerebral perfusion, thereby linking vascular dysfunction directly to neurodegenerative pathology [3,4,7,29].

#### 4.1.2. Alzheimer’s Disease

Insights from genome-wide association studies (GWASs) have underscored the critical involvement of microglial immune mechanisms in the pathophysiology of Alzheimer’s disease (AD). Several genes identified as risk alleles—most notably TREM2 and APOE—are predominantly expressed in microglial cells, suggesting these cells play a central role in disease susceptibility [49].

Contrary to earlier hypotheses that emphasized a classic pro-inflammatory microglial response, postmortem analyses of AD patients have revealed the presence of morphologically distinct microglial phenotypes surrounding amyloid-β (Aβ) deposits and tau aggregates. These cells exhibit features that align more closely with dystrophic or senescent microglia, rather than with the activated states typically observed in acute inflammatory contexts such as sepsis. This has led to the proposal that microglial dysfunction in AD contributes more to synaptic degradation and neuronal loss than to inflammatory damage per se [14,50].

In physiological contexts, microglia play a pivotal role during neurodevelopment, eliminating neurons that fail to integrate into functional networks and actively remodeling synaptic connections. This pruning process is mediated through the classical complement cascade, wherein synaptic targets are tagged with C1q and C3 proteins and subsequently recognized by complement receptor 3 (C3R) on microglia. The removal occurs via trogocytosis, a process that selectively targets presynaptic terminals and axonal segments. These mechanisms are particularly relevant to AD, where synaptic loss and cognitive decline accompany the accumulation of pathological protein aggregates, implicating aberrant microglial activity in disease progression [13].

Genetic and molecular risk factors for AD, such as APOE and TREM2 variants, not only drive neurodegeneration but also amplify BBB vulnerability, impairing clearance mechanisms and enhancing perivascular inflammation [14,49,50].

Microglial activation exerts profound effects on both the BBB and AD pathology. Pro-inflammatory cytokines (e.g., IL-1β, TNF-α) alter endothelial permeability, while microglia-derived MMPs degrade basement membrane components, destabilizing the barrier. At the same time, chronic activation promotes synaptic toxicity, accelerates tau hyperphosphorylation, and facilitates Aβ deposition. Thus, microglia act as central players at the interface between neurovascular dysfunction and neurodegeneration, reinforcing the concept that BBB breakdown and AD progression are interconnected processes [1,21,22,39].

### 4.2. Inflammation

#### 4.2.1. Blood–Brain Barrier

Neuroinflammatory responses in AD are frequently accompanied by alterations in the cerebral microvasculature. Notably, systemic infections that do not directly affect the CNS can still trigger brain inflammation, suggesting that peripheral immune activity contributes to the neuroinflammatory milieu. Although the BBB serves as a gatekeeper between the bloodstream and the CNS, endothelial cells are not passive participants. Instead, they actively transmit inflammatory signals to adjacent perivascular macrophages and microglia, which may in turn affect neurons and glial cells within the brain parenchyma.

Key pro-inflammatory cytokines—such as IL-6, TNF-α, and IL-1β—can traverse the BBB through saturable, receptor-mediated transport systems [29]. The efficiency of transport depends on both the molecular structure of the cytokine and its plasma concentration. Some cytokines are internalized by endothelial cells, while others penetrate deeper into the CNS tissue. Interestingly, studies have demonstrated that plasma from elderly individuals can enhance the expression of vascular cell adhesion molecule-1 (VCAM-1) in cerebral endothelial cells. This upregulation promotes microglial activation and facilitates the adhesion and migration of leukocytes, reflecting endothelial activation and immune infiltration [12,36,37].

From an immunological perspective, it is now well established that CNS inflammation increases the permeability of the BBB, especially in the presence of antigenic stimulation. This heightened permeability allows the infiltration of T lymphocytes and neutrophils into the brain, a hallmark of neuroinflammatory activity. The molecular pathways leading to this barrier disruption involve complex interactions between inflammatory mediators and perivascular cells—particularly microglia—which play a pivotal role in modulating endothelial integrity and facilitating immune cell transmigration [13].

#### 4.2.2. Alzheimer’s Disease

Data from postmortem analyses of Alzheimer’s disease (AD) brains—obtained through immunohistochemical, biochemical, and molecular techniques—have made it possible to classify the wide array of inflammatory molecules found within the central nervous system (CNS) of affected individuals. Central to the inflammatory cascade in AD are Aβ deposits, tau-based neurofibrillary tangles, and progressive neuronal degeneration—hallmark lesions that have been recognized for nearly a century [51]. Similar to peripheral tissues, where chronic injury and persistent exposure to aberrant materials trigger immune activation, the accumulation of Aβ and tau in the brain is believed to initiate a sustained inflammatory response. Once this is set in motion, a multitude of immune pathways become involved, with overlapping feedback loops and intricate cross-talk between mediators [35,39].

This complexity makes it likely that the activation of one inflammatory axis can influence many others. Therefore, any attempt to identify a primary driver of inflammation in AD is more a matter of convenience than definitive hierarchy. Whether focusing on complement activation, cytokine release, chemokine signaling, acute-phase reactants, or other pathways, current evidence suggests that no single mechanism outweighs the others in its contribution to disease progression [25,32,45,48].

Epidemiological studies have reported a decreased risk of developing AD in individuals chronically treated with anti-inflammatory medications for conditions such as arthritis. This supports the hypothesis that inflammation plays a significant role in AD pathology. Inflammation, as an innate immune reaction to injury, infection, or systemic disorders—including obesity—can be acute or chronic [43]. While acute inflammation presents with clear clinical signs such as fever, pain, and swelling, chronic inflammation often persists silently and may underlie neurodegenerative processes. In the AD brain, inflammatory cells and mediators accumulate in close proximity to Aβ plaques. Once initiated, the inflammatory cascade leads to the release of cytotoxic factors that compromise cellular membranes, promoting neuronal lysis and the spread of damage across broader regions of the brain [17,21].

Inflammation bridges vascular dysfunction and Alzheimer’s pathology. Cytokines such as IL-1β and TNF-α destabilize endothelial and pericytic interactions, increasing paracellular permeability, while systemic inflammation exacerbates BBB leakage. Simultaneously, these inflammatory cascades promote tau hyperphosphorylation and Aβ deposition, directly accelerating cognitive decline. Consequently, inflammatory signaling represents a unifying mechanism that links BBB breakdown with hallmark features of AD, demonstrating the inseparable nature of vascular and neurodegenerative processes [21,27,34].

### 4.3. Vascular Damage

#### 4.3.1. Blood–Brain Barrier

Alterations in cerebral blood flow in Alzheimer’s disease (AD) exacerbate vascular injury, a process further intensified by the breakdown of the blood–brain barrier (BBB). Emerging research attributes this deterioration, in part, to the degeneration of pericytes, contractile cells that enwrap the endothelium and contribute to BBB stability [11,15,28,35,52]. Neuroimaging has revealed that the hippocampus, a region essential for memory, is particularly susceptible to pericyte-related barrier dysfunction, with this vulnerability increasing as individuals age [8]. Transcriptomic analyses using VINE-seq (vessel isolation and nuclear extraction for sequencing) in human brain tissue have identified multiple vascular alterations in patients with AD compared to cognitively healthy controls [5,45]. The reduced number of recovered vascular cell nuclei in AD samples, validated by immunostaining, suggests cell loss within the cerebral vasculature. Additionally, significant changes in gene expression were observed in mural cells and fibroblasts, both of which play essential roles in maintaining vascular tone and barrier architecture. These molecular alterations imply impaired vasoconstriction and disrupted cerebral perfusion, potentially explaining some of the cerebrovascular abnormalities detected in AD [46].

Hypoxic and ischemic events are closely associated with the disruption of endothelial junctions that maintain BBB integrity. This breakdown is mediated by key molecular players, including inflammatory cytokines, vascular endothelial growth factor (VEGF), and nitric oxide (NO). In particular, elevated concentrations of IL-1β and TNF-α have been reported in such conditions. These mediators enhance the expression of adhesion molecules on endothelial cells, as well as on circulating immune cells like monocytes and neutrophils, facilitating their transmigration across the BBB. As a result, leukocyte infiltration into the brain parenchyma increases. While these inflammatory cascades contribute to barrier damage, astrocytes may act as modulators of injury, playing a protective role by regulating vascular responses during such insults [53].

#### 4.3.2. Alzheimer’s Disease

Vascular dysfunction in Alzheimer’s disease (AD) creates conditions where circulating blood components and neural proteins can interact in harmful ways [38]. One of the clearest examples of this pathological interface is cerebral amyloid angiopathy (CAA) [9,18], a condition marked by the deposition of Aβ within and around the cerebral vasculature. CAA is observed in approximately 80–95% of individuals with AD and is associated with endothelial injury, vascular instability, and inflammatory processes. As a consequence, patients with CAA present a higher risk of experiencing cerebrovascular events, particularly stroke, compared to those without vascular amyloid deposition [9,30]. Additionally, inherited variants of CAA have been linked to altered neural network connectivity, indicating that vascular compromise may directly contribute to cognitive decline [46]. The compromised vascular barrier in CAA enhances the interaction between Aβ and plasma proteins like fibrinogen, a key factor in coagulation. When Aβ binds to fibrinogen, it alters fibrin clot architecture and impairs fibrinolysis, thus exacerbating vascular dysfunction [43,54]. Specific mutations in Aβ, particularly those linked to early-onset AD, show stronger affinities for fibrinogen, further increasing amyloid accumulation within the vasculature and intensifying endothelial damage [6]. Compounding this, neuroinflammation has been identified as a major contributor to vascular and neural deterioration in AD. Autopsy reports have consistently shown elevated inflammatory markers [33], and neuroimaging has revealed persistent inflammation across the disease course [7,33,43].

Moreover, growing evidence supports a connection between cardiovascular disease and the risk of developing Alzheimer’s disease. As early as 1969, studies reported increased levels of homocysteine (Hcy)—a byproduct of protein metabolism—in the urine of patients with cardiovascular pathology. Elevated Hcy in the circulation can contribute to vascular damage by promoting the oxidation of cholesterol and compromising endothelial function [15]. These vascular insults may serve as a common mechanistic link between systemic cardiovascular risk factors and neurodegeneration observed in AD [10].

Vascular injury in Alzheimer’s disease is not limited to reduced cerebral perfusion, but extends to disruption of endothelial signaling and basement membrane integrity. Hypoperfusion promotes ischemia and oxidative stress, while impaired transport systems such as GLUT1 and LRP1 facilitate Aβ accumulation in the parenchyma. These findings illustrate that vascular damage acts as a central link between BBB breakdown and Alzheimer’s pathology, reinforcing that cerebrovascular dysfunction is both a consequence and a driver of neurodegeneration [18,21,23,27,40].

### 4.4. Oxidative Stress

#### 4.4.1. Blood–Brain Barrier

The degradation of epithelial and endothelial barriers is a key event in the development of inflammation following oxidative stress. Endothelial cells serve both as initiators and as targets of reactive oxygen species (ROS), which contribute to vascular injury by modifying lipids and proteins within affected cells [55]. This oxidative damage triggers the activation and release of inflammatory mediators, including cytokines and proteolytic enzymes, which in turn amplify tissue injury. In the context of acute ischemic stroke, the most extensively studied pro-inflammatory cytokines include TNF-α, various interleukins (IL-1β, IL-6, IL-10, IL-17, IL-20), and transforming growth factor-beta (TGF-β), all of which play pivotal roles in orchestrating vascular and immune responses [15,53]. Among these, interleukin-17A (IL-17A) has been shown to stimulate ROS production via NADPH oxidase or xanthine oxidase pathways. The resulting oxidative stress enhances the activity of endothelial contractile proteins and leads to reduced expression of occludin, a tight junction protein essential for blood–brain barrier integrity. Notably, inhibiting either ROS generation or myosin light-chain phosphorylation has been found to prevent IL-17A-induced disruption of the BBB [9,28,40,44].

#### 4.4.2. Alzheimer’s Disease

Oxidative stress is a widely recognized contributor to neurodegenerative conditions, including Alzheimer’s disease (AD). Both oxidative imbalance and the aggregation of Aβ oligomers are events that occur prior to the manifestation of clinical symptoms in AD. Research using transgenic mouse models carrying AD-related genetic mutations has consistently shown elevated levels of oxidative damage markers [11,19,27]. The brain is especially susceptible to oxidative injury due to several intrinsic factors: its high demand for oxygen, abundance of polyunsaturated fatty acids, limited antioxidant defenses, and the presence of transition metals capable of redox cycling [56]. The development of animal models expressing key mutations associated with AD has facilitated the investigation of oxidative stress in the early phases of the disease. Notably, biochemical signs of oxidative damage—particularly those linked to lipid peroxidation—can be detected before the formation of Aβ plaques and cognitive decline, making them promising indicators for early diagnosis [30]. Common biomarkers of oxidative damage include lipid peroxidation products such as F2-isoprostanes (F2-IsoPs), F4-neuroprostanes (F4-NeuroPs), malondialdehyde (MDA), and 4-hydroxy-2-nonenal (HNE) [7,13,32,37]. In addition, protein oxidation is evidenced by increased carbonylation, while oxidative injury to nucleic acids is reflected in elevated levels of 8-oxo-2′-deoxyguanosine (8-OHdG) and 8-hydroxyguanosine (8-OHG), which serve as key markers of DNA and RNA damage, respectively [48].

Oxidative stress not only compromises BBB endothelial integrity through downregulation of tight junction proteins and mitochondrial dysfunction but it also synergizes with Aβ toxicity to aggravate neuronal injury. Excessive ROS production facilitates Aβ aggregation, enhances tau phosphorylation, and impairs glucose metabolism, creating a vicious cycle between vascular and neurodegenerative damage. These findings demonstrate that oxidative stress is not simply a vascular stressor but an upstream driver that links BBB breakdown with Alzheimer’s disease pathology [4,21,27,40].


**In addition to inflammation, oxidative stress, vascular injury, and impaired amyloid-β clearance, other mechanisms reinforce the interconnection between blood–brain barrier (BBB) dysfunction and Alzheimer’s disease (AD) pathogenesis.**


-Pericyte degeneration: Pericytes are perivascular cells crucial for the stability and function of the blood–brain barrier (BBB). Their degeneration, observed in the early stages of AD, contributes to the loss of the blood–brain barrier, cerebral hypoperfusion, and impaired neuronal homeostasis [8,28]. Murine models have demonstrated that pericyte loss is directly associated with white matter damage and cognitive deficits [48].-Nutrient transport dysfunction: The glucose transporter GLUT1, which is essential for brain energy supply, is decreased in the blood–brain barrier (BBB) of patients with AD [27]. This metabolic dysfunction compromises neuronal viability, facilitates the accumulation of pathological proteins, and accelerates neurodegeneration [4,22,23]. The dysfunction of GLUT1 in AD not only impairs glucose transport but may also indirectly affect Aβ clearance. Reduced energy availability can compromise the function of ATP-dependent efflux transporters such as P-glycoprotein (P-gp), which is essential for Aβ removal. Moreover, GLUT1 deficiency may alter endothelial cell homeostasis and reduce the expression or activity of LRP1, another key transporter involved in Aβ transcytosis across the BBB. Therefore, GLUT1 impairment may exacerbate Aβ accumulation by both limiting metabolic support and impairing clearance mechanisms [7,33].-Loss of polarity in astrocytes: Astrocytes normally regulate metabolic exchange and blood–brain barrier (BBB) integrity through their perivascular terminals. In AD, the loss of polarity in astrocytes alters their supportive functions, increasing vascular permeability and exacerbating neuroinflammation [2,20].-Compromised lymphatic system: BBB dysfunction alters the glymphatic clearance system of the brain, reducing the clearance of pathological proteins such as Aβ and tau, and favoring the degenerative processes typical of AD [3,9].-Endothelial epigenetic modifications: Epigenetic alterations in brain endothelial cells, driven by chronic inflammation and aging, reduce the expression of essential tight junction proteins and transporters and promote the disruption of the blood–brain barrier (BBB) [33]. These modifications could represent an early link between environmental risk factors and susceptibility to AD [57].-Dysregulation of the Wnt/β-catenin pathway: The Wnt/β-catenin signaling pathway is crucial for the maintenance of the blood–brain barrier (BBB). Its inhibition, observed in AD, leads to decreased expression of tight junction proteins and structural weakening of the barrier [29].-Hormonal stress: Mild stress, chronically experienced, aggravates and accelerates the main features of the disease in people with a genetic predisposition to develop Alzheimer’s disease. Many studies have shown that stress can cause cognitive impairments. In addition, patients with depression experience episodes of memory loss, and stress is a factor associated with depression [14,34,52]. Stress and hormones released during stress exposure can also alter the normal functioning of the BBB, since most cells involved in BBB formation (endothelial cells, astrocytes, and microglia) have receptors for glucocorticoids, corticotropin-releasing hormone, and adrenaline. In adult mammals, acute stress modifies BBB permeability to circulating molecules in the blood, and several studies have reported an increase in BBB permeability after acute stress [7].

Most of the findings regarding stress-induced BBB permeability derive from models of acute stress, such as restraint stress or forced swim tests in rodents. However, chronic stress paradigms—particularly those simulating prolonged social or psychological stress—have also demonstrated BBB disruption and increased glucocorticoid receptor activation in endothelial and glial cells. These chronic models may better reflect real-world conditions associated with increased AD risk in humans [14,33].

These additional mechanisms reinforce the hypothesis that blood–brain barrier (BBB) dysfunction is not simply a secondary event but a fundamental and early contributor to the Alzheimer’s disease cascade. Therefore, therapeutic strategies aimed at preserving BBB integrity have the potential to delay or prevent neurodegeneration [44,50].

### 4.5. What Starts Outside the Organism (Parabiosis)

The Aβ protein performs numerous basic functions in the body. However, when they acquire the wrong structure, these proteins stick together, forming fibers that, in turn, aggregate into oligomers and Aβ plaques, which are highly toxic to neurons. Numerous studies have suggested that these plaques are responsible for brain cell death that triggers Alzheimer’s disease. However, this protein is not unique to the brain but is produced throughout the body [4,19]. Notably, alpha 1-antichymotrypsin has been shown to bind Aβ peptides in a sequence-specific manner and to modulate fibril disaggregation in vitro [31]. Therefore, is it possible that Aβ protein of “extracerebral” origin also contributes to the onset and progression of Alzheimer’s disease?

To answer this question, the authors resorted to a technique called “parabiosis,” in which two living beings are surgically joined to form, as if they were Siamese twins, a “single organism” that shares the physiological systems of its two predecessors. They used two mice and “stitched” them together to create a “single mouse” with a shared circulatory system. The first mouse was completely normal and, therefore, unable to develop Alzheimer’s disease. However, the second mouse was genetically modified to carry a mutation responsible for the production of high levels of Aβ protein. Thus, after several months, a large number of Aβ plaques were observed in the brain. At this point, he was stitched together with his healthy counterpart for one year so that the brains of both animals ended up sharing the disease [3].

While the parabiosis experiment is intriguing and highlights the potential systemic contribution of peripheral Aβ to Alzheimer’s disease (AD), it also raises important limitations that must be critically considered. First, this model involves surgically conjoining two mice to create a shared circulatory system—a biological condition that has no direct equivalent in humans. Consequently, the metabolic, immunological, and neurovascular interactions observed in these animals may not fully reflect the physiological dynamics of the human body [3,43].

Second, murine models inherently differ from humans in terms of lifespan, immune system complexity, blood–brain barrier architecture, and amyloid processing pathways. These interspecies differences limit the extent to which results can be extrapolated to human AD pathology. Furthermore, the genetically modified mouse used in this study produces abnormally high levels of Aβ, which may not replicate the slow, multifactorial nature of Aβ accumulation in sporadic human Alzheimer’s cases [9].

Nevertheless, the findings offer compelling evidence that peripheral sources of Aβ can cross into the brain and participate in plaque formation. This suggests that systemic factors—such as impaired peripheral clearance mechanisms, liver or kidney dysfunction, or chronic inflammation—could play a more substantial role in AD pathogenesis than previously thought. From a therapeutic perspective, this opens the door to novel approaches aimed at reducing peripheral Aβ levels, enhancing systemic clearance pathways, or even developing plasma-based exchange therapies. However, these strategies must first be rigorously validated in translational models that better mimic human pathophysiology [2,3].

## 5. Results

Recent findings support the notion that Alzheimer’s disease (AD) arises not solely from central nervous system dysfunction but also from systemic processes contributing to the accumulation of Aβ. This perspective opens up therapeutic opportunities that extend beyond the brain, such as peripheral interventions involving liver and kidney clearance of circulating Aβ through targeted molecules capable of binding and facilitating its elimination [14,50].

Our histological analysis of the blood–brain barrier (BBB) reinforces the hypothesis that increased permeability of this structure may represent a crucial early event in AD pathogenesis. The BBB is not simply a passive protective barrier but a dynamic interface highly sensitive to many of the same pathogenic factors implicated in AD—oxidative stress, inflammation, vascular dysfunction, and aging. However, a paradox emerges: while the BBB permits the entry of harmful agents under pathological conditions, it often prevents the passage of therapeutic compounds intended to restore brain homeostasis. This raises an urgent and unresolved question: What distinguishes the physicochemical properties of noxious substances that cross the BBB from those of drugs designed to benefit the brain? Is the selectivity of a damaged BBB altered in such a way that it favors pathology over repair? These are challenges that demand deeper investigation.

Advances in drug delivery strategies, including functionalized nanoparticles, have begun to overcome these obstacles by enabling targeted and controlled delivery of therapeutic agents across the BBB to sites of neurodegeneration [2]. Moreover, preventive strategies focused on restoring or resealing BBB integrity are gaining attention, as chronic exposure to blood-borne toxins is increasingly seen as a driver of neurodegeneration [57]. Stress, as a modifiable risk factor, may also influence both BBB permeability and AD vulnerability. While glucocorticoid-mediated stress responses vary among individuals, preclinical models suggest that those with higher stress resilience may exhibit resistance to AD pathology [15,53,57]. These findings suggest that the future of AD prevention and treatment lies not only in targeting neural circuits, but also in addressing systemic, vascular, immunological, and behavioral factors that converge at the blood–brain interface.

In addition to Aβ and tau pathologies, recent studies have highlighted the role of iron dysregulation in AD, especially its interplay with BBB dysfunction. Iron deposition in the brain has been associated with oxidative stress, neuroinflammation, and cognitive impairment. Disruption of the BBB may exacerbate iron accumulation by altering iron transport and storage mechanisms, as shown in recent neuroimaging and neuropathological studies. These findings underscore the multifactorial nature of BBB failure and its broader implications beyond Aβ clearance, encompassing other toxic elements such as free iron that can contribute to neurodegeneration [49,58].

## 6. Conclusions

The breakdown of the blood–brain barrier (BBB) is increasingly recognized not merely as a consequence but as a central, active contributor to the pathogenesis of Alzheimer’s disease (AD). Far from being a passive filter, the BBB plays a dynamic regulatory role in maintaining central nervous system (CNS) homeostasis through selective transport, immune surveillance, and neurovascular coupling. Its structural integrity—maintained by endothelial cells, pericytes, astrocytic endfeet, and extracellular matrix components—is progressively compromised in AD due to a multifactorial convergence of amyloid pathology, tau-related cytotoxicity, chronic neuroinflammation, oxidative stress, and vascular dysfunction.

The evidence reviewed suggests that BBB disruption precedes cognitive symptoms and facilitates a vicious cycle: impaired clearance of amyloid-β, increased infiltration of peripheral inflammatory mediators, dysregulated cerebral perfusion, and enhanced oxidative damage. Cerebral amyloid angiopathy, pericyte degeneration, tight junction downregulation, and the aberrant activation of microglia all converge at the interface between vascular and neurodegenerative processes. This intersection reveals that AD should be reconsidered, at least in part, as a neurovascular disorder.

Despite these advances, current therapeutic approaches largely overlook the vascular and barrier-related aspects of AD. Targeting BBB preservation—whether by modulating endothelial signaling, stabilizing pericyte function, inhibiting pro-inflammatory cascades, or restoring transporter systems—offers an underexplored but potentially transformative avenue for early intervention and disease modification. Future research should aim not only to characterize BBB dysfunction as a biomarker of early AD but also to exploit it as a therapeutic entry point. Understanding and restoring BBB function may ultimately prove essential for altering the course of neurodegeneration in Alzheimer’s disease.

## Figures and Tables

**Figure 1 neurolint-17-00166-f001:**
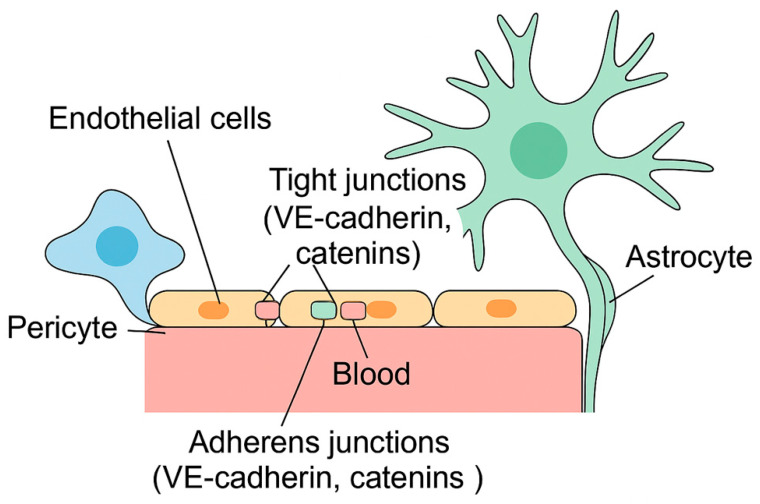
Schematic representation of the blood–brain barrier (BBB) and the neurovascular unit. The image illustrates the key cellular and molecular components involved in BBB function, including endothelial cells sealed by tight and adherens junctions, surrounding pericytes embedded within the basement membrane, and astrocytic endfeet enveloping the vascular surface. Together, these elements regulate selective permeability and maintain CNS homeostasis.

## Data Availability

Not applicable.

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
