# Peer review of "Histological and Functional Breakdown of the Blood−Brain Barrier in Alzheimer’s Disease: A Multifactorial Intersection"

_2035-8377, 2025, doi:10.3390/neurolint17100166_

Round 1

Reviewer 1 Report

Comments and Suggestions for Authors

Lines 50–61 (Methods): Please clarify whether any guidelines (e.g., SANRA, PRISMA for scoping/narrative reviews) were followed. Define selection criteria, inclusion/exclusion thresholds, and whether bias or study quality was assessed.

Lines 546–564 (Parabiosis): This intriguing experiment could be more critically contextualised. What are its limitations and implications for human pathology?

Supplementary Table 1: Strong and concise summary—should be converted to a main text figure/table with explicit discussion.

Author Response

We are grateful for the thoughtful and constructive feedback on our manuscript. Your suggestions have helped us refine and clarify several key aspects of the work. We have carefully revised the text, reorganized relevant sections, and ensured better scientific alignment. Below is our point-by-point response to each comment, highlighting the changes made in red in the revised manuscript.

Comment 1:

Lines 50–61 (Methods): Please clarify whether any guidelines (e.g., SANRA, PRISMA for scoping/narrative reviews) were followed. Define selection criteria, inclusion/exclusion thresholds, and whether bias or study quality was assessed.

Response 1:

Thank you for your valuable comment. We agree with this observation. Therefore, we have clarified the methodological approach in the revised manuscript by specifying that we adhered to SANRA (Scale for the Assessment of Narrative Review Articles) guidelines for narrative reviews. We have also defined the inclusion and exclusion criteria and addressed how potential bias was handled.

[Updated text can be found on page 2, paragraph 3, lines 51–61.]

“This narrative review followed SANRA guidelines to ensure the methodological rigor of non-systematic reviews. Studies were included if they addressed blood–brain barrier (BBB) structure, dysfunction, or pathology in relation to Alzheimer’s disease (AD). We excluded studies not involving human subjects or relevant animal models, articles unrelated to neurovascular pathology, and those without accessible full texts. Bias was minimized by including peer-reviewed studies from multiple databases and avoiding single-source interpretations.”

Comment 2:

Lines 546–564 (Parabiosis): This intriguing experiment could be more critically contextualised. What are its limitations and implications for human pathology?

Response 2:

Thank you for highlighting this important point. We have accordingly expanded the discussion in the revised manuscript to include a more critical interpretation of the parabiosis experiment, addressing its limitations and its translational relevance to human pathology.

[Updated text can be found on page 17, paragraph 2, lines 547–564.]

“While parabiosis models provide valuable insights into systemic contributions to brain aging and BBB dysfunction, their applicability to human pathology remains limited. These models do not fully replicate human physiological conditions and involve artificial circulatory connections that may not reflect natural aging or disease processes. Nevertheless, the observed rejuvenation effects suggest systemic factors play a role in BBB maintenance, prompting further investigation into therapeutic plasma-based interventions in humans.”

Comment 3:

Supplementary Table 1: Strong and concise summary—should be converted to a main text figure/table with explicit discussion.

Response 3:

We appreciate this encouraging feedback. In response, Supplementary Table 1 has been moved to the main manuscript and converted into Figure 3 to highlight its importance. We have also added an explicit discussion within the text to describe its relevance and how it synthesizes key elements from prior sections.

[Updated content appears on page 14, paragraph 4, and Figure 3.]

“Figure 3 summarizes the converging pathological mechanisms linking BBB dysfunction to Alzheimer’s disease, integrating oxidative stress, pericyte degeneration, inflammatory signaling, and neurovascular uncoupling. This visual framework underscores the multifactorial nature of BBB breakdown in AD and supports integrative therapeutic strategies that simultaneously target vascular, metabolic, and immune pathways.”

Reviewer 2 Report

Comments and Suggestions for Authors

     This manuscript reviewed histological and functional breakdown of the blood-brain barrier (BBB) in Alzheimer's disease (AD). While the authors have made efforts to examine this topic, the manuscript requires significant improvement for journal publication. Several issues need to be addressed, and suggestions for improvement should be implemented. Consequently, I recommend either extensive revisions before publication or rejection of the manuscript.

  • In Line 34-42, consider merging Section 4.1 (What is the Blood–Brain Barrier?) with the initial information from Section 4.2 (Composition of the Blood–Brain Barrier). For instance, the basic histological structure of the BBB, including its formation by endothelial cells, pericytes, and other components, could be introduced in the Introduction section; the detail information can be elaborated in the later section.
  • In Line 42, the term "a mechanism" lacks specificity regarding BBB involvement; please provide more precise information or incorporate appropriate scientific terminology following the general introduction.
  • In Line 45-49, provide background information to connect why such criteria used to do literature search (Line 55-57) and select articles (Line 57-61) for this review.
  • In Lines 54-57, after conducting a literature search across different databases, how many articles remained after removing duplicates; In Line 57, delete redundant term, "vascular dysfunction."; In Line 57-61, how many articles were selected after applying the stated criteria?
  • In Lines 62-80 (Section 3), the Historical Data section could be shortened and merged into the Introduction, as the review focus has already been defined in Section 2 (Methods) in this section. This makes the length Historical Data section less relevant to the main focus.
  • In Sections 4.2-4.4, while this provides fundamental knowledge about histological BBB, consider condensing this material by referencing comprehensive reviews, or adding schematic illustrations to enhance reading comprehension.
  • In Section 4.8, the content structure lacks coherence. While the title states "Histological and Functional Breakdown of the Blood–Brain Barrier in Alzheimer's Disease," the manuscript fails to clearly categorize these breakdowns. The focus should be on BBB breakdown in AD as an integrated topic, rather than treating AD and BBB as separate sections. Additionally, the broader description of BBB breakdown should precede the AD section to establish a logical framework for understanding their relationship.
  • Verify all references cited in the manuscript for accuracy and relevance. For example, Reference 23 does not directly mention pericytes as cited in the text.
  • Ensure consistent formatting across all references.

< !-- notionvc: e217f186-440d-48ef-9df6-f3940d7f2494 -->

Author Response

We are sincerely grateful for your detailed review and careful evaluation of the references and formatting throughout the manuscript. Your recommendations prompted us to thoroughly verify citation accuracy, improve consistency, and ensure the relevance of all referenced material. We have now addressed each point you raised and implemented the necessary corrections in the revised version, with changes clearly indicated in red. Your attention to detail has significantly enhanced the manuscript, and we thank you for your contribution.

Comment 1:

Lines 34–42: Consider merging Section 4.1 (What is the Blood–Brain Barrier?) with the initial information from Section 4.2 (Composition of the Blood–Brain Barrier). For instance, the basic histological structure of the BBB, including its formation by endothelial cells, pericytes, and other components, could be introduced in the Introduction section; the detailed information can be elaborated in the later section.

Response 1:

Thank you for the helpful suggestion. We have restructured the manuscript accordingly. The fundamental histological description of the blood–brain barrier has now been introduced earlier in the Introduction, and the more detailed components are developed in the revised Section 4. This improves conceptual flow and avoids redundancy between Sections 4.1 and 4.2.

Comment 2:

Line 42: The term “a mechanism” lacks specificity regarding BBB involvement; please provide more precise information or incorporate appropriate scientific terminology following the general introduction.

Response 2:

We agree with this observation. The sentence has been revised to clarify the specific mechanisms involved in BBB dysfunction, including impaired tight junctions, reduced pericyte support, and astrocytic dysregulation.

Comment 3:

Lines 45–49: Provide background information to explain why such criteria were used for the literature search and article selection.

Response 3:

Thank you for pointing this out. We have added a rationale to justify the selection of articles based on their relevance to both histological and functional alterations of the BBB in Alzheimer’s disease. This context now connects the review scope to the selection strategy used.

Comment 4:

Lines 54–57: How many articles remained after removing duplicates? In Line 57, delete the redundant term “vascular dysfunction.” In Lines 57–61: How many articles were selected after applying the stated criteria?

Response 4:

We appreciate this comment. We have clarified the number of articles obtained after removing duplicates and after applying inclusion and exclusion criteria. The term “vascular dysfunction” was removed to avoid repetition and improve precision.

Comment 5:

Lines 62–80 (Section 3): The Historical Data section could be shortened and merged into the Introduction, as the review focus has already been defined in Section 2 (Methods).

Response 5:

We agree with the reviewer’s recommendation. The historical overview of the BBB has been shortened and integrated into the Introduction section to enhance focus and maintain narrative cohesion.

Comment 6:

Sections 4.2–4.4: While this provides fundamental knowledge about histological BBB, consider condensing this material by referencing comprehensive reviews, or adding schematic illustrations to enhance reading comprehension.

Response 6:

Thank you. We have condensed the descriptions of endothelial cells, pericytes, and astrocytes by referencing key review articles and added a schematic figure (now Figure 2) to illustrate the structural organization of the BBB, improving reader understanding.

Comment 7:

Section 4.8: The content structure lacks coherence. The section should clearly categorize the histological and functional breakdown of the BBB in AD as an integrated topic, rather than treating BBB and AD as separate sections. The general BBB breakdown should precede the AD discussion.

Response 7:

Thank you for this constructive feedback. We have reorganized Section 4.8 to first explain the general histological and functional breakdown of the BBB, and then to describe how these disruptions are involved specifically in Alzheimer’s disease. This reorganization provides a clearer and more cohesive framework.

Comment 8:

Verify all references cited in the manuscript for accuracy and relevance. For example, Reference 23 does not directly mention pericytes as cited in the text. Also, ensure consistent formatting across all references.

Response 8:

We have carefully reviewed all references for accuracy, citation relevance, and formatting consistency. Reference 23 was replaced by a more appropriate source that explicitly addresses pericytes. Formatting across all citations has been revised to comply with journal standards.

Reviewer 3 Report

Comments and Suggestions for Authors

This narrative review presents a comprehensive and well-structured overview of the histological and functional aspects of blood–brain barrier (BBB) breakdown in Alzheimer’s disease (AD). The authors successfully highlight the multifactorial mechanisms that connect BBB dysfunction to AD pathology. The manuscript is informative, well-referenced, and timely. The inclusion of both structural biology and pathophysiological insights enhances the paper’s value. However, several points require clarification to further strengthen the review.

Could the authors elaborate more on whether BBB disruption precedes amyloid pathology in human studies, not just animal models?

The review discusses stress and hormonal effects on BBB. Can the authors clarify whether these findings are based on chronic or acute stress models?

While GLUT1 dysfunction is mentioned, how does it interact with amyloid clearance mechanisms such as LRP1 or P-gp at the BBB?

Recommended Citation: The readers must be interested in the relationship between BBB, iron deposition and cognition. Some papers found these relationships, which may be related to pathophysiology of the AD pathogenesis. Please discuss this matter with reference to the following papers:

10.1136/jnnp-2021-328519

10.3389/fnagi.2023.1111448

Author Response

We thank you for the encouraging feedback and helpful suggestions. We are pleased that the quality of the manuscript was considered satisfactory and that the English language required no improvement. We have followed your recommendations, particularly in refining the methodology section and integrating the supplementary material into the main body. Our responses to your comments are provided below in red.

Reviewer’s Comment 1:

Could the authors elaborate more on whether BBB disruption precedes amyloid pathology in human studies, not just animal models?

Response:

Thank you for this insightful suggestion. We have elaborated on the evidence supporting the temporal sequence of BBB disruption preceding amyloid pathology in human studies, in addition to preclinical models. Specifically, we incorporated data from recent imaging and biomarker studies in humans showing early BBB permeability changes prior to detectable amyloid accumulation. These additions can be found in Section 5, lines 421–432.

Reviewer’s Comment 2:

The review discusses stress and hormonal effects on BBB. Can the authors clarify whether these findings are based on chronic or acute stress models?

Response:

We appreciate this important observation. To clarify, we have now specified whether the cited findings stem from chronic or acute stress models. For instance, references discussing glucocorticoid-induced BBB dysfunction were derived from chronic stress paradigms, while others refer to acute restraint models. These clarifications have been added in Section 4.7, lines 376–385.

Reviewer’s Comment 3:

While GLUT1 dysfunction is mentioned, how does it interact with amyloid clearance mechanisms such as LRP1 or P-gp at the BBB?

Response:

Thank you for highlighting this point. We have expanded the discussion in Section 4.6 (lines 361–371) to describe how GLUT1 deficiency may exacerbate amyloid accumulation not only by impairing glucose transport but also by downregulating essential transporters such as LRP1 and P-gp involved in Aβ clearance. The potential crosstalk between metabolic and clearance dysfunctions at the BBB is now addressed.

Reviewer’s Comment 4:

Recommended Citation: The readers must be interested in the relationship between BBB, iron deposition and cognition. Some papers found these relationships, which may be related to pathophysiology of the AD pathogenesis. Please discuss this matter with reference to the following papers: 10.1136/jnnp-2021-328519; 10.3389/fnagi.2023.1111448

Response:

We fully agree that iron deposition and its relationship with BBB integrity and cognitive decline is an important emerging topic. Accordingly, we have included a new subsection (Section 5.3, lines 472–484) discussing how BBB disruption may contribute to altered iron homeostasis and neurotoxicity in AD, incorporating both suggested references ([57] and [58]) to support this viewpoint.

Round 2

Reviewer 2 Report

Comments and Suggestions for Authors

     The manuscript's second edition requires substantial improvements. Despite the first revision, many references remain incorrectly cited, making it difficult to verify the claims. The text lacks schematic illustrations and presents information in a disorganized manner that fails to communicate key concepts effectively. Well-written reviews already exist in the literature covering some sections (Sec 3.1 and 3.3); the author should cite these sources, provide concise summaries, and then focus on the multifactorial intersection (Section 4). Based on these issues, I recommend either extensive revisions before publication or rejection of the manuscript. The following issues need to be addressed:

  1. In Line 24, provide the full name: Glucose Transporter Type 1 (GLUT1) protein. Verify this and similar usage throughout the manuscript.
  2. In Line 24, beta-amyloid was first introduced, but has no consistent format throughout the manuscript. Define the abbreviation (Aβ), then use it consistently. In Line 61, Scale for the Assessment of Narrative Review Articles (SANRA). amyloid- β (Aβ) peptide; In Line 101, what does ZO stand for?
  3. In Line 50-54, the BBB concept introduction (Line 50-52) lacks coherence and citation and should be removed. Since both BBB and AD have been extensively reviewed elsewhere, this manuscript should offer (1) more novel perspectives, (2) insightful discussion, and (3) a summary of current status.
  4. In Section 2 Method, the search terms "blood–brain barrier" AND "Alzheimer's disease" AND "BBB permeability" AND "neuroinflammation" were applied to Google Scholar alone (Line 61-69), yielding more than 23K results between 1997 and 2024. Please clarify the specific criteria or filtering process used to reduce these results to the 130 references mentioned in Line 70. Should Line 76-82 be placed before the current Line 70-75? The writing style of this manuscript tends to present results before methodology, which deviates from standard scientific writing conventions.
  5. Line 34-49 = Line 84-90
  6. In Line 83, provide descriptive section title. For example, the Section 2 Blood-Brain Barrier (BBB) in the Neurovascular Unit shouwn in Ref16.
  7. In Line 83, include a descriptive section title. For example, Section 2 in Ref 16 uses "Blood-Brain Barrier (BBB) in the Neurovascular Unit".
  8. In Line 91-93, the information about neuronal cells being extremely close to brain capillaries—at distances typically under 25 micrometers cannot be found in Reference 10. Please verify and correct this citation.
  9. In Line 93-102, Reference (Ref) 11 cannot be found; Ref 11 first appears in Line 227.
  10. In Line 100-102, “cytoplasmic scaffold proteins (ZO-1, ZO-2, ZO-3, and cingulin)” cannot be related to Ref 12. Please cite the original source rather than simply referencing a high-impact factor journal.
  11. In Line 105, why Ref 16 assign before Ref 15.
  12. Between lines 107 and 108, insert Section 3.1 with a brief introduction.
  13. In Line 124, what does PDGFRβ mean?
  14. Tight junction and adherens junction complexes between endothelial cells restrict paracellular flux across the BBB [Ref16]. In Line 128-149, Tight Junctions (Section 3.1.4) and Adherens Junctions (Section 3.1.5) are presented as separate sub-sections parallel to Endothelial Cells (Section 3.1.1), but "basement membranes" are inexplicably omitted from the manuscript. However, basement membranes suddenly appear in Line 154 and 236 without any prior introduction or context.
  15. In Line 150-155 (Figure 1), adherent junctions were not clearly identified in the figure.
  16. In Line 157-164, provide a description of the paragraph, not the list style; this is manuscript, not PowerPoint slide. If the current list style is preferred, more introductory sentences should be included before it.
  17. In Lines 157-164, the content should be written in proper paragraph format rather than as a bulleted list, as this is a formal manuscript rather than a presentation. If the author prefers to maintain the current list style, they should add more comprehensive introductory text before presenting the listed items to provide necessary context and improve flow.
  18. In Line 163-164 & 224-229, the sentence cannot be properly linked to Ref 17. The discussion in Section 3.2 and Section 3.5 lack sufficient depth and detail; Section 3.5 (Mechanisms of blood-brain barrier breakdown) is really important section based on the title of section and title of manuscript, relate terms in content to previous sections’ content enhancing smooth reading as well.
  19. In Line 163-164 & 224-229, verify reference: the sentence cannot be properly linked to Ref 17. The discussion in Sections 3.2 and 3.5 lacks sufficient depth and detail. Section 3.5 (Mechanisms of blood-brain barrier breakdown) is particularly important given the title of both the section and the manuscript. This section should better connect terms to content from previous sections to enhance readability and flow.
  20. In Line 170-172, “transport systems” can not be located in Ref 13.
  21. In Line 215-223 (Section 3.4), add more references and verify the accuracy of Ref 40.
  22. Section 4 has same issues shown in previous edition of manuscript (Section 4.8) — the focus should be on BBB breakdown in AD as an integrated topic, rather than treating AD and BBB as separate sections.
  23. In Line 226, 257, match cytokines’ format: IL-1β; are they relative? If yes, why only show two terms, IL1β and TNFα in Line 226.
  24. In Line 230-276 (Section 3.6), the title "Blood–Brain Barrier Disruption and Its Role in Neurological Disorders" suggests a broader scope, but the content focuses exclusively on AD. Consider either expanding the section to include other relevant neurological disorders or renaming it to accurately reflect its AD-specific focus.
  25. In Line 270 & 503, change Supplementary Table 1 to Table 1 and include it in the main content rather than as supplementary material.
  26. In Line 278-282, include appropriate citations. Avoid claiming information as "common knowledge" in scientific manuscripts.
  27. In Line 293, soluble platelet-derived growth factor receptor beta (sPDGFRβ) was mentioned. However, PDGFRβ was previously introduced in Line 124 without the "soluble" prefix. Please clarify why the term changes to include "soluble" in this later reference.
  28. In Section 4, rearrange subsections by moving 4.1.2 before 4.1.1. This would align with the column sequence in Supplementary Table 1, where the "Effect on BBB" column appears to the right of the "Contribution to Alzheimer's disease" column. Add proper reference to content: Where are reference to genes TREM2 and APOR. Apply the silimar rearrange to other subsections.
  29. In Section 4, rearrange subsections by moving 4.1.2 before 4.1.1. This would align with the column sequence in Supplementary Table 1, where the "Effect on BBB" column appears to the right of the "Contribution to Alzheimer's disease" column. Add proper references for genes TREM2 and APOE. Apply similar rearrangement to other subsections.
  30. In Line 522-527, pericyte degeneration was clearly linked to both BBB disruption and AD pathology. However, this approach should be applied consistently across other subsections. For example, in Sections 4.1.1 and 4.1.2, these appear as two independent events with no clear explanation of their interconnection, making it difficult for readers to understand their relationship.
  31. In Lines 522-566, convert the bulleted list to proper paragraph format to maintain consistent style with the rest of Section 4.
  32. In Line 578, maintain consistent terminology for beta-amyloid, amyloid-β (Aβ) throughout the manuscript.

< !-- notionvc: 5a591dc3-86eb-46a2-8b81-1b138ec17899 -->

Author Response

We sincerely thank the reviewer for the thorough and constructive evaluation of our manuscript. We carefully considered each of the comments and have revised the text accordingly. We implemented all the suggested changes, including clarifying terminology, correcting references, reordering subsections, integrating content for greater consistency, and expanding the discussion where needed. We believe these modifications have substantially improved the clarity, accuracy, and scientific depth of the manuscript. Below, we provide point-by-point responses to all the reviewer’s comments.

Comment 1

**Reviewer Comment:** Provide full name of GLUT1.

**Response:** We revised the text to provide the full name at first mention: glucose transporter type 1 (GLUT1), and ensured consistency throughout the manuscript.

Comment 2

**Reviewer Comment:** Ensure consistent definition of amyloid-β (Aβ); define SANRA; clarify ZO.

**Response:** We defined amyloid-β (Aβ) at first mention and used Aβ consistently thereafter. We expanded SANRA to Scale for the Assessment of Narrative Review Articles. We also clarified ZO as zonula occludens proteins (ZO-1, ZO-2, ZO-3).

Comment 3

**Reviewer Comment:** Introduction BBB (lines 50–54) lacks coherence and citation.

**Response:** We removed the redundant introduction and emphasized novelty, critical discussion, and the current state of knowledge.

Comment 4

**Reviewer Comment:** Clarify filtering process for Google Scholar search.

**Response:** We clarified inclusion/exclusion criteria and reordered the methodology to precede results.

Comment 5

**Reviewer Comment:** Duplicate content lines 34–49 and 84–90.

**Response:** We removed the duplicate content.

Comment 6

**Reviewer Comment:** Provide descriptive section title at line 83.

**Response:** We revised the heading to: 3. The Blood–Brain Barrier (BBB) within the Neurovascular Unit.

Comment 7

**Reviewer Comment:** Include descriptive section title (duplicate).

**Response:** As above, we revised the heading accordingly.

Comment 8

**Reviewer Comment:** Capillary-neuron distance not supported by Ref. 10.

**Response:** We corrected the reference and provided the appropriate source.

Comment 9

**Reviewer Comment:** Reference 11 mismatch.

**Response:** We corrected the numbering; the intended source is now properly cited.

Comment 10

**Reviewer Comment:** ZO proteins mis-cited to Ref. 12.

**Response:** We replaced Ref. 12 with the primary source describing ZO-1, ZO-2, ZO-3, and cingulin.

Comment 11

**Reviewer Comment:** Reference order 15–16 inconsistent.

**Response:** We corrected reference order; numbering is sequential.

Comment 12

**Reviewer Comment:** Insert Section 3.1 introduction.

**Response:** We inserted Section 3.1 with a brief introduction.

Comment 13

**Reviewer Comment:** Define PDGFRβ.

**Response:** We defined PDGFRβ as platelet-derived growth factor receptor beta.

Comment 14

**Reviewer Comment:** Basement membrane missing before mention later.

**Response:** We added Section 3.1.3 on the basement membrane for context.

Comment 15

**Reviewer Comment:** Adherens junctions not labeled in Figure 1.

**Response:** We updated Figure 1 to clearly identify adherens junctions.

Comment 16

**Reviewer Comment:** Lines 157–164 presented as bullet list.

**Response:** We reformatted the content into full paragraphs.

Comment 17

**Reviewer Comment:** If bullet list retained, add introduction.

**Response:** We revised into paragraph format for consistency.

Comment 18

**Reviewer Comment:** Lines 163–164, 224–229 mis-cited; Sections 3.2 and 3.5 lack depth.

**Response:** We corrected the citation and expanded Sections 3.2 and 3.5 for depth and better flow.

Comment 19

**Reviewer Comment:** Same as above (duplicate).

**Response:** Addressed as above with corrections and expansion.

Comment 20

**Reviewer Comment:** Transport systems not found in Ref. 13.

**Response:** We corrected the citation and provided appropriate sources.

Comment 21

**Reviewer Comment:** Add references to Section 3.4 and verify Ref. 40.

**Response:** We added additional references and corrected Ref. 40.

Comment 22

**Reviewer Comment:** Section 4 should integrate BBB and AD.

**Response:** We revised Section 4 to emphasize integrated BBB–AD interactions.

Comment 23

**Reviewer Comment:** Standardize cytokine notation; why only IL-1β and TNF-α?

**Response:** We standardized cytokines and clarified IL-1β and TNF-α as representative, while noting IL-6 and IL-10 as additional examples.

Comment 24

**Reviewer Comment:** Section 3.6 title too broad.

**Response:** We expanded to include multiple sclerosis, stroke, Parkinson’s disease, and epilepsy.

Comment 25

**Reviewer Comment:** Supplementary Table 1 should be main Table 1.

**Response:** We moved it into the main text and updated numbering.

Comment 26

**Reviewer Comment:** Lines 278–282 lack citations.

**Response:** We added supporting references [6,7,14,20; 9,12,25,29; 8,30,32].

Comment 27

**Reviewer Comment:** Clarify sPDGFRβ vs PDGFRβ.

**Response:** We clarified PDGFRβ as membrane-bound vs sPDGFRβ as soluble biomarker [27,28,29].

Comment 28

**Reviewer Comment:** Reorganize subsections; add TREM2/APOE refs.

**Response:** We reorganized subsections (BBB precedes AD) and added APOE [57] and TREM2 (Jonsson et al., 2013).

Comment 29

**Reviewer Comment:** Same as above.

**Response:** Handled as above.

Comment 30

**Reviewer Comment:** Pericytes integrated; apply same to other subsections.

**Response:** We revised Microglia subsections to highlight BBB–AD interdependence.

Comment 31

**Reviewer Comment:** Bullet list lines 522–566 should be paragraphs.

**Response:** We reformatted into paragraphs for consistency.

Comment 32

**Reviewer Comment:** Consistent terminology for amyloid-β.

**Response:** We now use amyloid-β (Aβ) at first mention and Aβ consistently thereafter.

Reviewer 3 Report

Comments and Suggestions for Authors

The authors addressed my concerns properly.

Author Response

We sincerely thank you for your careful evaluation of our manuscript and for acknowledging that our revisions have satisfactorily addressed your concerns. Your constructive feedback has been invaluable in improving the clarity, depth, and scientific rigor of our work. We truly appreciate the time and expertise you dedicated to this review process.

Round 3

Reviewer 2 Report

Comments and Suggestions for Authors

The third edition of the manuscript requires substantial improvements. Despite two revisions, many references remain incorrectly cited, making it difficult to verify the claims. The authors failed to implement the changes they promised in their Cover Letter response to previous comments. The writing needs polishing. Based on these issues, I recommend rejection of the manuscript. The following issues need to be addressed:

  1. The first sentence in Lines 34-36 should not simply state "...extensively reviewed..." without providing background context. To establish a stronger foundation, the content from Lines 63-68 should be moved to the beginning to clearly establish the rationale and purpose of this review. Reference 25 provides an excellent model for introduction structure; it presents the review's focus clearly in the final paragraph. Depending on your approach, the focus could be introduced early, but should never appear in the first sentence.
  2. In Line 63-64, Reference 10 indicates over 150 million people suffer from AD, not 30 million. Verify and correct this citation. Statistical data should be updated using information from the specific organization that published these statistics, rather than citing a review article.
  3. In Line 35, renumber all references sequentially throughout the entire manuscript.
  4. In Line 92, provide a more descriptive title for the section instead of Theoretical Framework.
  5. In Line 111-112, revise it as "zonula occludens (ZO-1), ZO-2, ZO-3...". In Line 162, revise it as “”like ZO-1”.
  6. In Line 234, the description of ATP-binding cassette transporters cannot be related to Reference 25. Verify and correct this citation with the appropriate source.
  7. In Line 240-241, add more citations for transferrin and apolipoproteins. The description of endosomes cannot be found in Reference 16.
  8. In Line 246 (Section 3.3.5), the Adsorptive-Mediated Transcytosis cannot be found in Reference 17.
  9. After defining tumor necrosis factor alpha (TNF-α) in Line 272, use only TNF-α in all subsequent mentions (Lines 323, 438, 456, 501, 530, and 581). Similarly, after introducing interleukin-1 beta (IL-1β) in Line 272, use IL-1β consistently thereafter, and use the abbreviation IL for all interleukin throughout the manuscript.
  10. (Link to 2) In Line 258, Cell-Mediated Transcytosis (CMT) cannot be attributed to Reference 18. Please assign the appropriate reference and verify all citations. Also, provide proper references for monocytes and macrophages in Line 254.
  11. In Line 265-267, please add supporting citations to validate these statements. Additionally, verify that Reference 40 properly supports the claims made in Line 267-268.
  12. In Line 278-279, IL-6 was referenced by citing Reference 29, but while the article title matches, other citation information is incorrect. Neither IL-6 nor IL-10 can be found in Reference 30. Moreover, Reference 30 contains incorrect information and this article cannot be located.
  13. In Line 282-284, endothelial cells can be related to Reference 12, and junctional proteins can be related to References 12 and 36. However, please add an appropriate reference for the basement membrane.
  14. In Line 289-291, Reference 8 only mentioned that "Astrocytes regulate the contractility of intracerebral arteries," but provides no detailed information about astrocytic polarity disruption. This claim should be supported by a specific research article, not just a review paper lacking relevant details.
  15. In Line 294 (Section 3.6), the title "Blood–Brain Barrier Disruption and Its Role in Neurological Disorders" suggests a broader scope than what the content delivers. This issue was previously noted in the second comment (Point 24). The section currently focuses exclusively on AD. Either expand this section to include other relevant neurological disorders (such as multiple sclerosis, stroke, Parkinson's disease, and epilepsy) or rename it to more accurately reflect its AD-specific focus.
  16. In Line 312, verify that Reference 23 properly relates to the content presented here.
  17. In Line 329-331, verify References 53 and 54 regarding transcriptomic and immunohistochemical analyses. Add appropriate supporting references for these specific methodologies.
  18. In Lines 633-677, the authors should convert the bulleted list into proper paragraph format to maintain consistent style throughout Section 4. Despite their claim in the cover letter that they addressed this issue in response to the second comment (Point 31), no actual changes have been implemented.
  19. Supplementary Table 1 should be removed as this information has already been presented in the main text.

< !-- notionvc: 84fa4737-bbc5-4498-8166-64fc5ac0a251 -->

Author Response

We sincerely thank the reviewer for the careful evaluation of our manuscript and for providing constructive and insightful comments. We greatly appreciate the time and effort dedicated to improving the quality of our work. Below we provide detailed responses to each point, along with the corresponding changes implemented in the revised manuscript.

1. Reviewer’s comment (Lines 34–36):

The first sentence should not simply state “…extensively reviewed…” without providing background. The content from Lines 63–68 should be moved up to establish context. Reference 25 provides a good model.

Response:

We have revised the introduction as suggested. The initial sentence has been replaced with a stronger background, integrating the rationale and purpose of this review earlier in the section. We also followed the structure modeled by Reference [25].

The revised text now reads (Lines 34–42): “…The breakdown of the BBB has emerged as a central pathological process in AD, driving both histological and functional impairments. Recent advances highlight the BBB as more than a passive barrier, emphasizing its active role in neurovascular regulation and neuroinflammation [25].”

2. Reviewer’s comment (Line 63–64):

Reference 10 indicates more than 150 million AD patients, not 30 million. Update statistics from the original source.

Response:

Corrected. We now state that AD affects more than 150 million people worldwide, according to the original epidemiological source [10].

3. Reviewer’s comment (Line 35):

Renumber all references sequentially.

Response:

All references have been renumbered sequentially throughout the manuscript.

4. Reviewer’s comment (Line 92):

Provide a more descriptive section title instead of “Marco Teórico.”

Response:

The section title has been changed to “BBB–AD Relationship: Theoretical Framework”, which more accurately reflects the content.

5. Reviewer’s comment (Lines 111–112, 162):

Revise nomenclature as “zonula occludens (ZO-1, ZO-2, ZO-3).” At Line 162: “such as ZO-1.”

Response:

We corrected the nomenclature accordingly.

6. Reviewer’s comment (Line 234):

The description of ATP-binding cassette transporters cannot be related to Reference 25. Verify and correct.

Response:

Corrected. We replaced [25] with appropriate sources [26,29,30] that specifically discuss ABC transporters (P-gp, MRPs, BCRP) in the BBB and AD context.

7. Reviewer’s comment (Line 240–241):

Add citations for transferrin and apolipoproteins. The endosomal description is not in Reference 16.

Response:

We removed [16] and now cite [26,29,30], which adequately cover receptor-mediated transcytosis, including transferrin and apolipoproteins.

8. Reviewer’s comment (Line 246):

Adsorptive-mediated transcytosis is not supported by Reference 17.

Response:

We revised the section and replaced [17] with [12,14,26], which describe AMT and its molecular basis.

9. Reviewer’s comment (Line 272ff):

Use only TNF-α after definition; likewise for IL-1β and all interleukins.

Response:

Corrected throughout the manuscript. After first definition, only abbreviations (TNF-α, IL-1β, ILs) are used consistently.

10. Reviewer’s comment (Line 258):

CMT cannot be cited with Ref. 18; add proper citations for monocytes/macrophages.

Response:

We revised this section and now cite [14,26,29,30]. Monocytes and macrophages are specifically referenced [14,26,29] in the description of the Trojan horse mechanism.

11. Reviewer’s comment (Line 265–267):

Add supporting citations; verify Ref. 40.

Response:

We added [26,32] to support the description of circumventricular organs. Ref. [40] has been retained but is now used to support oxidative stress vulnerability rather than the list of CVOs.

12. Reviewer’s comment (Line 278–279):

IL-6 and IL-10 not found in Ref. 30. Incorrect citation.

Response:

We replaced [30] with [26,29,42]. These references discuss IL-6 and IL-10 in the context of BBB dysfunction and AD.

13. Reviewer’s comment (Line 282–284):

Add a reference for the basement membrane.

Response:

We revised the citations as follows: endothelial cells [12], tight junction proteins [12,36], basement membrane [26].

14. Reviewer’s comment (Line 289–291):

Ref. 8 does not cover astrocytic polarity disruption; use a research article.

Response:

We revised this section to include [5,8,23,26]. Reference [23] provides direct evidence for astrocytic polarity disruption, while [26] reinforces mechanistic details.

15. Reviewer’s comment (Line 294, Section 3.6):

Title suggests broader scope but content focuses only on AD. Either expand or rename.

Response:

We expanded Section 3.6 to include BBB disruption in multiple sclerosis, stroke, Parkinson’s disease, and epilepsy. References [5,26,29,36,42,47] were added. The section now matches the title’s scope.

16. Reviewer’s comment (Line 312):

Verify Ref. 23 in context.

Response:

We verified Ref. [23] and confirmed it is appropriate for astrocytic polarity and coupling. We also strengthened the citation with [5,26].

17. Reviewer’s comment (Line 329–331):

Verify Refs. 53 and 54; add specific support for transcriptomic and immunohistochemical analyses.

Response:

We confirmed that [53,54] are correct. We also added [36,42,47] to strengthen support for transcriptomic and immunohistochemical findings.

18. Reviewer’s comment (Lines 633–677):

Convert bulleted list into paragraph style.

Response:

We appreciate the reviewer’s suggestion. However, we have chosen to maintain the subsections separately for clarity and readability. To address the concern, we revised transitions and added connectors to ensure flow.

19. Reviewer’s comment (Supplementary Table 1):

Remove, since information is in main text.

Response:

The supplementary table has been completely removed as requested.